# Paper-based ELISA diagnosis technology for human brucellosis based on a multiepitope fusion protein

Dehui Yin[1,☯], Qiongqiong Bai[1,☯], Xiling Wu[1], Han Li[2], Jihong Shao[1], Mingjun Sun[3,*], Hai Jiang[4,*], Jingpeng Zhang[1,*]

**1** Key Lab of Environment and Health, School of Public Health, Xuzhou Medical University, Xuzhou, China, **2** Department of Infection Control, the First Hospital of Jilin University, Changchun, China, **3** Laboratory of Zoonoses, China Animal Health and Epidemiology Center, Qingdao, China, **4** State Key Laboratory for Infectious Disease Prevention and Control, Chinese Center for Disease Control and Prevention, Beijing, China

☯ These authors contributed equally to this work.
* sunmingjun@cahec.cn (MS); jianghai@icdc.cn (HJ); xiaopangpeng@126.com (JZ)

## Abstract

### Background

Brucellosis, as a serious zoonotic infectious disease, has been recognized as a re-emerging disease in the developing countries worldwide. In china, the incidence of brucellosis is increasing each year, seriously threatening the health of humans as well as animal populations. Despite a quite number of diagnostic methods currently being used for brucellosis, innovative technologies are still needed for its rapid and accurate diagnosis, especially in area where traditional diagnostic is unavailable.

### Methodology/Principal findings

In this study, a total of 22 B cell linear epitopes were predicted from five *Brucella* outer membrane proteins (OMPs) using an immunoinformatic approach. These epitopes were then chemically synthesized, and with the method of indirect ELISA (iELISA), each of them displayed a certain degree of capability in identifying human brucellosis positive sera. Subsequently, a fusion protein consisting of the 22 predicted epitopes was prokaryotically expressed and used as diagnostic antigen in a newly established brucellosis testing method, nano-ZnO modified paper-based ELISA (nano-p-ELISA). According to the verifying test using a collection of sera collected from brucellosis and non-brucellosis patients, the sensitivity and specificity of multiepitope based nano-p-ELISA were 92.38% and 98.35% respectively. The positive predictive value was 98.26% and the negative predictive value was 91.67%. The multiepitope based fusion protein also displayed significantly higher specificity than *Brucella* lipopolysaccharide (LPS) antigen.

### Conclusions

B cell epitopes are important candidates for serologically testing brucellosis. Multiepitope fusion protein based nano-p-ELISA displayed significantly sensitivity and specificity

**Data Availability Statement:** All relevant data are within the manuscript and its Supporting information files.

**Funding:** This work was supported by Young Scientists Fund of the National Natural Science Foundation of China (Grant number 81802101, received by DY). The funders had no role in study design, data collection and analysis, decision to publish, or preparation of the manuscript.

compared to *Brucella* LPS antigen. The strategy applied in this study will be helpful to develop rapid and accurate diagnostic method for brucellosis in human as well as animal populations.

## Author summary

Brucellosis is one of the most important zoonosis in the world and has caused tremendous economic losses in agriculture and animal husbandry in many countries. Developing rapid, sensitive and specific diagnostic methods is very important for early detection and treatment of brucellosis patients. In this study, a novel diagnostic technique, nano-ZnO modified paper ELISA, was established. The antigen used in this technique was a fusion protein containing multiple B cell epitopes, which were predicted from *Brucella* major outer membrane proteins such as Bp26, Omp31, Omp16, Omp2b and Omp25. Comparing to traditional LPS antigen, this multiepitope based antigen displayed considerably higher sensitivity and higher specificity in laboratory. With the strategy described in this paper, more efficient epitopes and protein antigen can be identified in the future. Currently, LPS antigen is only prepared from live *Brucella*, while protein antigen can be produced in large quantities in prokaryotic expression system. In addition to nano-p-ELISA, this protein antigen can also be used for development other methods such as fluorescent polarization assay (FPA) and immunochromatographic assay (ICA) to meet the varied demand for brucellosis testing.

## Introduction

Brucellosis is a reemerging zoonotic infectious disease. It not only seriously threatens the health of the people but also causes huge economic losses to animal husbandry industry. In human, brucellosis often manifests multiple symptoms and a long course of disease. So it is often misdiagnosed and causes increased cost of treatment and waste of medical resources [1]. Therefore, a rapid and accurate testing technology is very important for brucellosis diagnosis and subsequent treatment.

 Currently, diagnostic methods for brucellosis include bacterial isolation, specific antibody detection and amplification of specific DNA fragments(PCR or qPCR) [2,3]. Bacterial isolation needs biosafety level 3 laboratory, and usually takes several weeks[4]. PCR or qPCR are fast and having higher sensitivity and specificity, but nucleic acid contamination often causes false positive result so that expensive facilities must be needed to guarantee the accuracy of testing [5]. On the other hand, antibody detection is the most popularly used method for testing brucellosis as they are easy to handle and suitable for most laboratories. There are several serological methods popularly used for antibody detection, including the agglutination test, complement fixation test (CFT), enzyme-linked immunosorbent assay (ELISA), immunochromatographic assay (ICA) and fluorescence polarization assay (FPA)[6]. All these methods are based on detecting antibody targeting for lipopolysaccharide (LPS). Although LPS is a major immunogen arousing high level of antibody titer, it contains the common epitope with other gram negative bacillus like *Yersinia* O:9 and *Escherichia* O:157, which greatly reduces its specificity in testing brucellosis. Moreover, as LPS is obtained only through culturing *Brucella* in high-level biosafety facilities, it is not available to most of diagnostic kit manufacturers. Thus,

seeking for new antigen to replace LPS is critical for developing easily available brucellosis testing kits.

In our previous study, the out membrane proteins such as Bp26 and Omp31 demonstrated considerable efficacy in detecting human brucellosis sera[7]. In this study, the B cell epitopes from five out membrane proteins were predicted and synthesized for the purpose of designing a more effective antigen. Paper-based enzyme-linked immunosorbent assay (p-ELISA) is an emerging technology. Due to the small reagent required and special equipment independence, it has attracted increasing attention from diagnostic reagent developers[8–10]. Here, using a prokaryotically expressed protein consisting of multiple B cell epitopes, a nano-ZnO modified p-ELISA (nano-p-ELISA) was established. The efficiency of this new method in detecting human brucellosis was evaluated against a collection of human sera. Hopefully, the epitope based protein can be applied in development of other fast and low cost diagnostic methods for brucellosis.

## Methods

### Ethics statement

All experiments were approved by the Ethics Committee of Xuzhou Medical University.

### Human serum samples

121 human brucellosis sera were gifted by the School of Public Health of Jilin University. 90 negative control sera were collected by Infection Department of the First Clinical Hospital of Jilin University, including 50 sera from healthy individuals and 40 patient sera confirmed by blood culture to be infected with other pathogens (S1 Table).

### B cell epitope prediction and synthesis

The amino acid sequences of *Brucella* out membrane protein Bp26, Omp2b, Omp16, Omp25 and Omp31 were download from NCBI website (https://www.ncbi.nlm.nih.gov/protein/). *Brucella* species and protein accession numbers were listed in supplementary S2 Table. The conserved amino acid sequences of these proteins were used to predict B cell epitopes using BepiPred tool in IEDB (http://tools.iedb.org/bcell/). Prediction threshold is 0.350(default value), above this threshold is possible epitope. Peptides longer than 6 amino acids were assumed as effective epitope and selected. Each of selected B cell epitope was chemically synthesized and coupled to keyhole limpet hemocyanin (KLH) in Sangon Biotech Company (Shanghai, China). The purity of each polypeptide-KLH was more than 90%.

### Evaluation of B cell epitopes

121 human brucellosis positive sera were used to verify the diagnostic effect of predicted epitopes by indirect ELISA (iELISA). Each peptide-KLH was diluted with carbonate buffer (pH = 9.6) to final concentrations of 30 μg/mL. 100 μL of peptide-KLH was added to 96-well plate (Corning, USA) and incubated overnight at 4°C. 300 μL blocking solution (5% skimmed milk in PBS) was then added to plate and incubated at 37°C for 1 h. After washing 3 times with PBST, 100 μL of 1:400 diluted serum was added and incubated at 37°C for 1 h. After washing 3 times with PBST, 1:5000 diluted HRP-labeled protein G (Thermo, USA) was added to plate and incubated at room temperature for 30 min. In the coloring step, 100 μL of TMB substrate solution was added to each well and incubated for 15 min at room temperature. Coloring was terminated by adding 50 μL of stopping solution (2 M $H_2SO_4$). The optical density was measured at 450 nm ($OD_{450}$) using ELISA plate reader (BioTek, USA). At the same time,

KLH (30 μg/mL, sigma) and lipopolysaccharide (LPS, 1 μg/mL provided by China Animal Health and Epidemiology Center) were used as controls in this experiment.

## Fusion protein preparation and verification

Selected peptides were concatenated together and adjacent peptides were connected by a 'GGGS' linker (S1 Fig). The DNA fragment corresponding to full length of concatenated peptides was synthesized and cloned into the expression vector pET-21a(+). Fusion protein containing concatenated peptides was expressed and purified from *E. coli* BL21(DE3) cells according to the optimized procedures (Sangon Biotech). The specific steps are described below.

After transferring the recombinant plasmid into BL21(DE3), 800 μL of nonresistant LB medium was added, followed by incubation at 37˚C for 45 min and centrifugation at 5000 rpm for 3 min. Most of the supernatant was discarded (leave approximately 100–150 μL), the bacteria were resuspended, the LB plate with corresponding resistance was selected, and it was coated. After air-drying, it was inverted and cultured overnight in a 37˚C incubator. The monoclonal colonies on the plate were chosen, placed into 10 mL of LB liquid medium and incubated at 37˚C and 200 rpm. The cultured bacterial solution was transferred to 750 mL of LB liquid medium at 37˚C and 200 rpm, cultured to $OD_{600}$ = 0.6–0.8 with IPTG (0.5 mM) at 16˚C and induced overnight. Then, the cells were centrifuged at 6000 rpm for 5 min, the supernatant was discarded, and the bacteria were collected. Bacteria were blown away with 20–30 mL 10 mM Tris-HCl (pH = 8.0) solution and ultrasonically broken (500 W, 60 times, 10 s each time, 15 s interval). After sonication, 100 μL of the bacterial suspension was centrifuged at 12000 rpm for 10 min, and 50 μL of supernatant was transferred to another EP tube. After the supernatant was removed, the precipitate was blown away with 50 μL of 10 mM Tris-HCl (pH = 8.0) solution. SDS-PAGE and Western blotting were used to detect protein expression. A nickel column (Ni Sepharose 6 Fast Flow, GE Healthcare) for affinity chromatography was used for protein purification. Taking 5 mL of Ni-NTA, the equilibrium column was washed with 5 times the column bed volume of binding buffer at a flow rate of 5 mL/min. The crude protein was incubated with the equilibrated column packing for 1 h; the incubated product was loaded onto the column and the effluent liquid was collected; the equilibrium column was washed with binding buffer; the column was washed with washing buffer, and the effluent liquid was collected; with the column was eluted with elution buffer, and the effluent liquid was collected; and the crude protein was treated, washed with effluent and eluted with effluent separately, followed by sample preparation and, SDS-PAGE and WB detection. The concentrated protein was divided into 1 mL/tube and stored at -80˚C.

## Evaluation of the diagnostic effect of fusion protein

Diagnostic effect of the fusion protein was evaluated according to the iELISA method described in **Evaluation of B cell epitopes section**. In this experiment, 1 μg fusion protein was coated to each well in 96-well plate, while other conditions were not changed.

## Establishment of nano-p-ELISA

ZnO nanorods were synthesized on Whatman No. 1 filter paper by a hydrothermal method [11]. Whatman filter paper was soaked in 100 mM zinc acetate solution for 60 s and then annealed at 100˚C for 1 h to form a seed layer. Then, the filter paper was transferred to a hydrothermal reaction vessel containing 100 mM hexamethylenetetramine and Zn$(NO_3)_2$.6H$_2$O. Whatman filter paper was left at 90˚C for 5 h for formation of ZnO nanorods. Filter paper was then immersed in an hydrous toluene solution with 1% APTES for 5 min,

dried at 100˚C for 15 min and then silanized. Scanning electron microscopy (SEM, JSM-7500F), X-ray diffraction (XRD, Bruker D8) and X-ray photoelectron spectroscopy (XPS, Escalable250Xi) were used to evaluate the nanorods structure on the surface of the paper. Nano-ZnO modified Whatman filter paper was punched into circular pieces with a diameter of 10 mm and A4 plastic packaging paper was punched into small holes with a diameter of 6 mm. The 10 mm circle paper was placed in the center of the 6 mm holes of the plastic packaging paper, fixed by a plastic packaging machine, and cut into small strips for further use.

### Evaluation of the diagnostic effect of nano-p-ELISA

Five microliters of fusion protein solution (30 μg/mL in PBS) was placed in each well, followed by incubation at room temperature for 30 min, washing with 20 μL of deionized water 3 times, and blocking with 20 μL of 5% skimmed milk powder at room temperature for 15 min. After washing 3 times with PBST, 5 μL of serum was added (diluted with 1:400) to the paper and incubated for 30 min. After washing 3 times with PBST, 5 μL of HRP labeled protein G was added (diluted with 1:8000), followed by incubation at room temperature for 210 s. After washing another 3 times, 5 μL of TMB substrate solution was added and incubated for 10 min. HP Laser Jet Pro MFP M227 was used to scan the paper. Image J software was used to carry out intensity analysis on developed color. To compare with the nano-p-ELISA method, the traditional p-ELISA (tra-p-ELISA) method was also performed according to literature[12]. Five microliters of chitosan dissolved in deionized water (0.25 mg/mL) was placed onto Whatman No.1 filter paper and air dried at room temperature. Then, 5 μL of 2.5% glutaraldehyde solution was added to the paper and incubated at room temperature for 2 h. The remaining steps are same as described in nano-p-ELISA.

### Statistical analysis

Dot plot and receiver operating characteristic (ROC) curve analyses were performed using GraphPad Prism version 6.05. The gray intensity were determined by Student's t-test (unpaired t-test). $P$-values$< 0.05$ were considered to be significantly different.

## Results

### B cell epitope prediction and evaluation

From five *Brucella* antigen proteins, BP26, Omp16, Omp25, Omp31 and Omp2b predicted 14, 8, 14, 12, 19 epitopes respectively, and finally a total of 22 epitopes were selected (Table 1). The length of these peptides ranged from 9 to 28 amino acid. The 22 polypeptide epitopes were synthesized and coupled to KLH. The results of iELISA showed that these epitopes have different ability in identifying human brucellosis positive sera (Fig 1). Average $OD_{450}$ value of KLH plus 3 times Standard Deviation (SD) was used as threshold to distinguish positive and negative samples, here the threshold value being calculated as 0.5178, the epitope with the highest capability was P19266-6, as from 121 human brucellosis serum samples, 79 samples were detected as above the threshold. The other six epitopes (P19266-5, 12, 13, 15, 16 and 18) displayed medium capability with 37, 43, 44, 33, 24 and 33 samples were identified respectively. The remaining epitopes showed only limited capability, with no more than 20 positive samples being detected (Table 1).

### Fusion protein preparation and its diagnostic effect

As each of 22 predicted epitopes demonstrated some extent discerning capability for brucellosis sera, all these epitopes were included for constructing a fusion protein. SDS-PAGE showed

**Table 1. Information about 22 predicted B cell epitopes.**

| Protein | Epitope (amino acid sequence) | Start-end position | Length | Peptide ID | Number of positive sera recognized (n = 121) |
|---|---|---|---|---|---|
| BP26 | AFAQENQMTTQPARIAV | 26–42 | 17 | P19266-1 | 18 |
| | KAGIEDRDLQTGGIN | 88–102 | 15 | P19266-2 | 17 |
| | QPIYVYPDDKNNLKEPTITGY | 104–124 | 21 | P19266-3 | 17 |
| | GVNQGGDLNLVNDNPSAVIN | 151–170 | 20 | P19266-4 | 19 |
| | LSRPPMPMP | 204–212 | 9 | P19266-5 | 37 |
| | AAAPDNSVPIAAGENSYNVSVNVVFE | 223–248 | 26 | P19266-6 | 79 |
| Omp2b | SGAQAADAIVAPEPEAVEY | 31–49 | 19 | P19266-7 | 10 |
| | DVKGGDDVYSGTDRNGWDK | 79–97 | 19 | P19266-8 | 12 |
| | NNSGVDGKYGNETSSGTV | 129–146 | 18 | P19266-9 | 10 |
| | TVTPEVSYTKFGGEWKNTVAEDNAWGGI | 341–368 | 28 | P19266-10 | 11 |
| Omp16 | AAAPGSSQDFTV | 44–55 | 12 | P19266-11 | 1 |
| | SRGVPTNRMRTISYGNERPVAVCD | 125–148 | 24 | P19266-12 | 42 |
| Omp25 | GRAKLENRTNGGTS | 56–69 | 14 | P19266-13 | 44 |
| | GNPVQTTGETQ | 115–125 | 11 | P19266-14 | 1 |
| | GGIKNSLRIGGEESSKSKTQT | 154–174 | 21 | P19266-15 | 33 |
| | GWTVGAGIEYAA | 175–186 | 12 | P19266-16 | 24 |
| | TDYGKKNFGLNDLDTRGSFKTNDIR | 199–223 | 25 | P19266-17 | 6 |
| Omp31 | VSEPSAPTAAPVDTFSWTGGYIGINA | 24–49 | 26 | P19266-18 | 33 |
| | GKFKHPFSSFDKEDNEQVSGSL | 53–75 | 23 | P19266-19 | 8 |
| | TGSISAGASGLEGKAE | 112–127 | 16 | P19266-20 | 7 |
| | GDDASALHTWSDKTKAGWTLGAGAEYA | 168–194 | 27 | P19266-21 | 4 |
| | DLGKRNLVD | 209–217 | 9 | P19266-22 | 8 |

that the molecular weight of prokaryotically expressed fusion protein was approximately 66 kDa (Fig 2A and 2B). Western blotting using anti-his tag antibody showed the same result (Fig 2C). Further mass spectrometry verified that the fusion protein was correctly expressed.

The diagnostic effect of the purified fusion protein was verified using121 human brucellosis sera and 90 control sera. For the fusion protein, the area under the ROC curve was 0.9877 (95% CI: 0.9758 to 0.9996), while the area under the ROC curve for LPS was 0.9174 (95% CI: 0.8796 to 0.9552) (Fig 3), indicating that fusion protein has higher diagnostic effectiveness than LPS. The optimal cutoff value was also calculated by the Youden index, under which the positive predictive value (PPV) and negative predictive value (NPV) of fusion protein were higher than those of LPS (Table 2). Under the cutoff value, 3 negative samples were misdiagnosed as positive with fusion protein. However, 20 negative samples was misdiagnosed as positive with LPS. This data indicated that the fusion protein had better specificity than LPS.

## Characterization of nano-ZnO

XRD results showed that the main composition of nano-ZnO was successfully formed (Fig 4A). Scanning electron microscopy showed the shape of nano-ZnO on the surface of Whatman filter paper (Fig 4B). XPS showed that the concentration of Zn atoms was 40.79% and the concentration of oxygen atoms was 59.21%, which further indicated that the nano-crystal was composed of ZnO.

## Evaluation of the diagnostic effect of nano-p-ELISA

Using 211 human brucellosis positive and negative sera, the ROC curve was obtained for nano-p-ELISA. Under the optimal cutoff value, 113 out of 121 positive samples were accurately

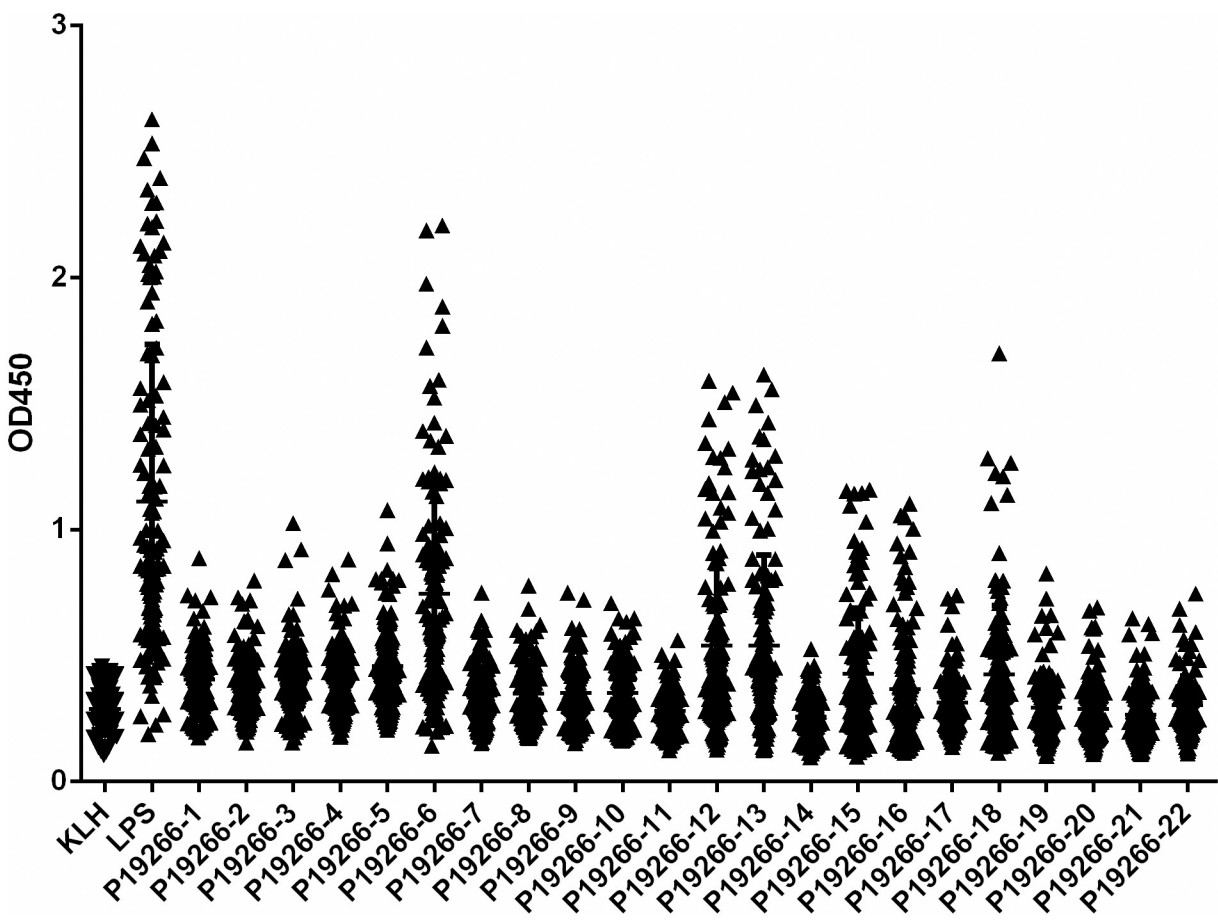

**Fig 1. Dotplot result of 22 epitopes in identifying 121 human brucellosis positive serum by iELISA.**

diagnosed, and 88 out of the 90 negative samples were correctly identified. The positive predictive value of the nano-p-ELISA was 98.26%, and the negative predictive value was 91.67% (Table 2). There was a significant difference between the positive and negative samples (*P*<0.001) (Fig 5A). The area under the curve was 0.9900 (95% CI, 0.9816 to 0.9984), indicating that this method performed well in diagnosing human brucellosis. The optimal cutoff

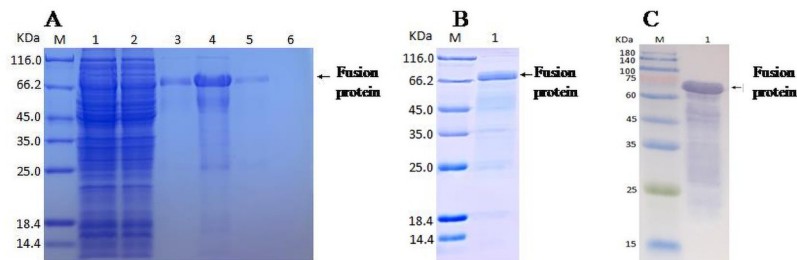

**Fig 2. SDS-PAGE and Western blot analysis of fusion protein.** (A)SDS-PAGE result of fusion protein in the process of purification (M, marker; lane1, loading solution; lane2, flow-through solution; lane3-4, 20 mM imidazole elution fraction; lane5, 50 mM imidazole elution fraction; lane6, 500 mM imidazole elution fraction). (B) SDS-PAGE result of purified fusion protein (M, marker; lane1, purified protein). (C) Western blot result of purified fusion protein (M, marker; lane1, purified protein).

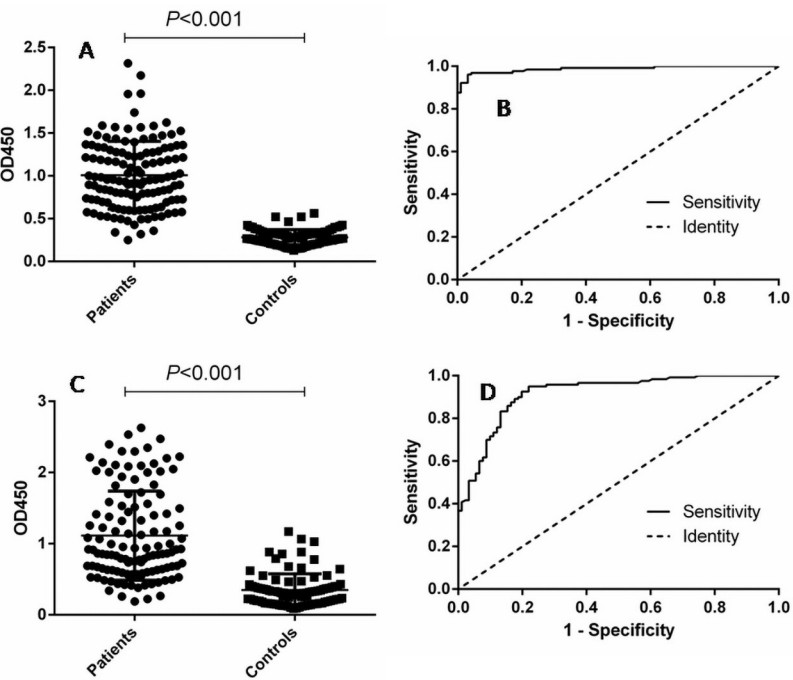

**Fig 3. Comparison of fusion protein and LPS in detecting human brucellosis positive and negative sera.** (A) Dotplot result of iELISA assay with fusion protein. (B) ROC analysis of iELISA assay with fusion protein. (C) Dotplot result of iELISA assay with LPS antigen. (D) ROC analysis of iELISA assay with LPS antigen.

value was 50.98, under which the sensitivity of this method was 92.38% (95% CI, 0.8554 to 0.9665) and the specificity was 98.35% (95% CI, 0.9416 to 0.9980) (Fig 5B).

Correspondingly, the gray intensities and ROC curve of tra-p-ELISA demonstrated a similar performance as nano-p-ELISA (Fig 5C and 5D). Sensitivity and specificity analysis also supported this conclusion (Table 2).

## Discussion

At present, the most of commercially available brucellosis detecting kits are based on the LPS which is polysaccharides compound on the surface of *Brucella*. Although single antigenic epitope of LPS can be chemically synthesized[13], under most circumstances, acquisition of LPS is only achieved by culturing live *Brucella* in high-level biosafety laboratories, which is an insurmountable obstacle for most diagnostics manufacturers. Therefore, seeking for new

**Table 2. Positive and negative predictive values of the test calculated for different cutoff values.**

| Cutoff value | Positive | | Negative | | PPV (%) | NPV (%) |
|---|---|---|---|---|---|---|
| | TP | FN | TN | FP | | |
| ≥0.470 (fusion protein) | 117 | 4 | 87 | 3 | 95.90 | 95.51 |
| ≥0.4095 (LPS) | 115 | 6 | 70 | 20 | 85.19 | 92.10 |
| ≥50.98 (nano-p-ELISA) | 113 | 8 | 88 | 2 | 98.26 | 91.67 |
| ≥45.66 (tra-p-ELISA) | 113 | 8 | 87 | 3 | 97.41 | 91.58 |

TP, true positives; TN, true negatives; FP, false positives; FN, false negatives; PPV, positive predictive value (TP/TP+FP)×100; NPV, negative predictive value (TN/TN+FN)×100.

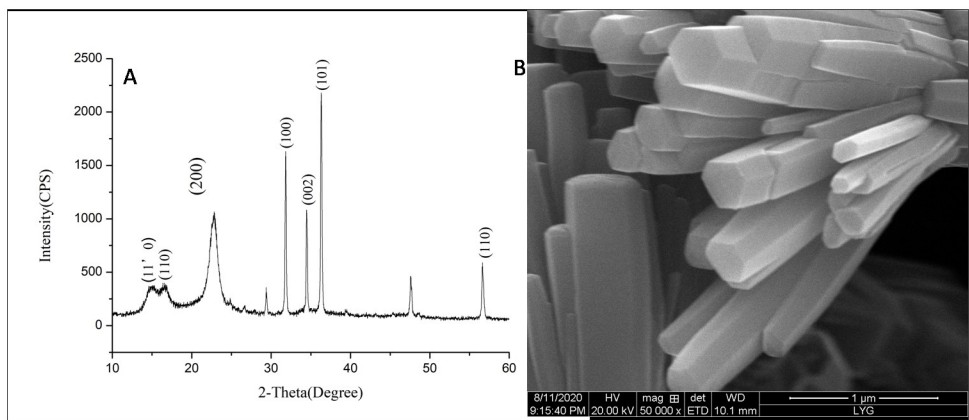

**Fig 4. Characterization of prepared nano-ZnO.** (A) XRD analysis of nano-ZnO. (B) The shape of nano-ZnO acquired by scanning electron microscope.

candidate antigen to replace LPS is of great significance for developing brucellosis detecting kits which can be easily produced and available to all users.

According to literatures, many *Brucella* OMPs manifested strong capacity in arousing humoral immune response [14–16], and some of these proteins have also been used for development of subunit vaccine against brucellosis[17,18]. In our previous study, some *Brucella* OMPs showed quite satisfactory result in diagnosing brucellosis, basically comparable to LPS antigen [7]. The data in this paper reflected that not only entire OMPs, the linear short peptides in these OMPs also maintain good antigenicity. Moreover, multiple epitopes were more effective than single epitopes in identifying human brucellosis positive sera. It seems that higher sensitivity of epitope-based protein antigen could be achieved by increasing the number of epitopes. But, more epitopes could imply lower specificity as the chance of including common epitopes with other pathogens can be simultaneously increased. Therefore, there is still a lot of trimming work to be done on this fusion protein, so that the most suitable epitopes are selected for future commercial use.

The immunoinformatic analysis is an emerging science that integrates life sciences, computer science and mathematics to accelerate the process for vaccine design, disease diagnosis and treatment, as well as diagnostic agent screening[19,20,21]. The online immunoinformatic tool (Bepipred Linear Epitope Prediction) used in this study were proved to be feasible in discerning viable B cell epitopes, as a quite number of predicted epitopes were subsequently confirmed to be effective in laboratory. On the other hand, there were still many predicted epitopes just showing limited diagnostic effects. Hopefully in the future, a combination of immunoinformatic tools can be set up and used to improve the efficiency of epitope prediction.

Fast and easy-to-perform are the most concerned features in developing novel brucellosis detection technologies, especially in the point-of-care testing of medical diagnosis, environmental surveillance and food safety analysis. P-ELISA has attracted the attention from many researchers due to its higher specificity, simplicity, rapidity, portability and low cost [22,23]. Currently, nanomaterial modified p-ELISA are more widely used as modification can increase the surface area of the paper[10,24–26]. Although tra-p-ELISA and nano-p-ELISA demonstrated similar antigenic capability, the latter method is more suitable for rapid on-site detection as it uses fewer reagent and can be stored at room temperature for a long time.

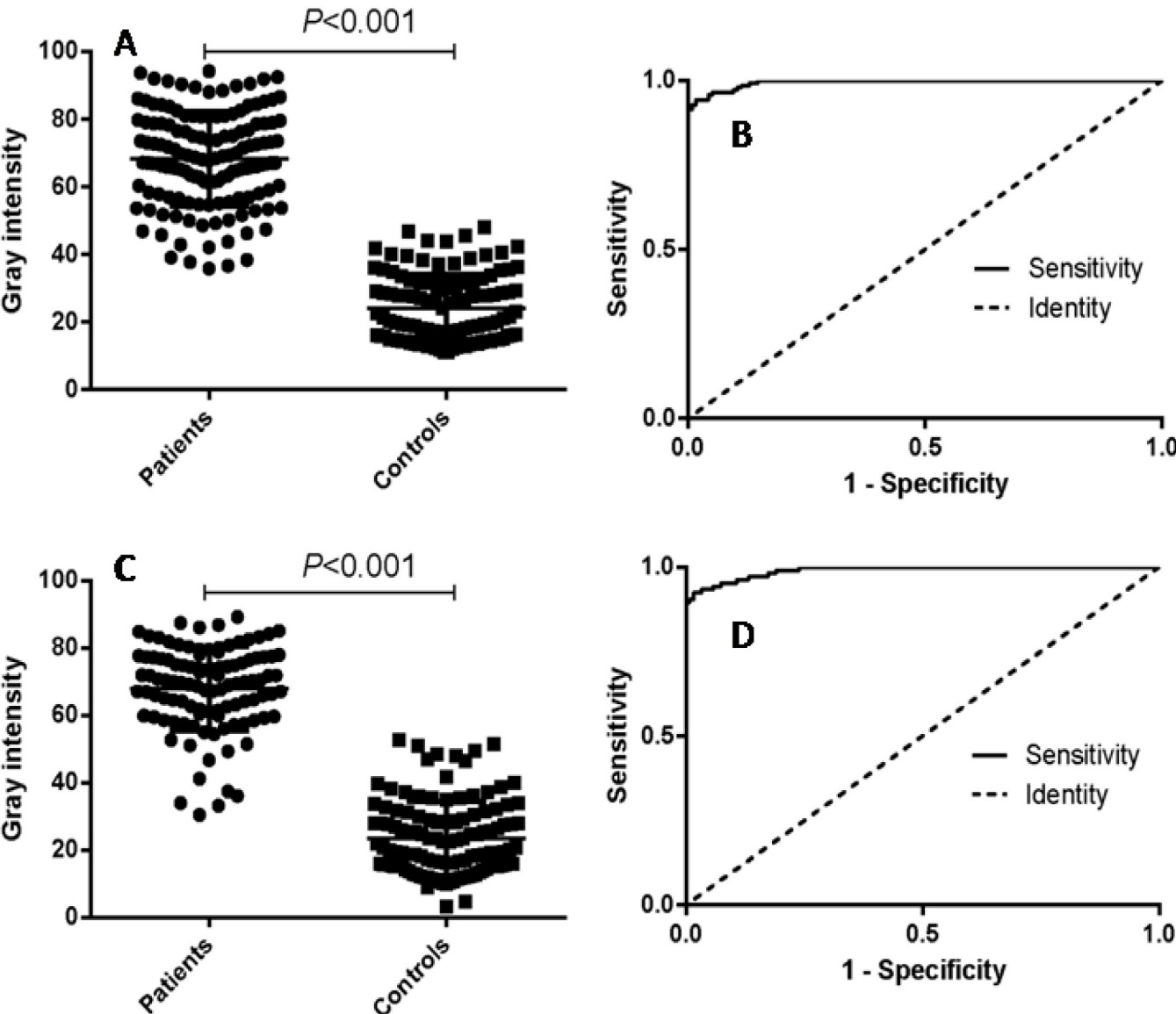

**Fig 5. P-ELISA analysis of human serum samples.** (A) Dotplotresult of the nano-p-ELISA assay. (B) ROC analysis of nano-p-ELISA assay results. (**C**) Dotplot result of tra-p-ELISA assay. (D) ROC analysis of tra-p-ELISA assay results.

In summary, using bioinformatic technology combined with nanomaterials, this performance has established a new type of brucellosis diagnostic technology, which has good potential application value. However, the brucellosis sera selected in this study were all clinically screened positive sera, and the number was limited. The diagnostic validity of this method requires a large number of clinical random samples for verification. Beside nano-p-ELISA, there are some other methods suitable for rapid on-site testing, such as FPA and ICA. Combining the multiepitope based antigen and well established fast testing methods, more brucellosis testing kits would be produced in the future to meet the varied demand for brucellosis testing.

## Supporting information

**S1 Fig. Comprehensive sequence of amino acids of the fusion protein; the linkers are in red font.**
(TIF)

**S2 Fig. Results of p-ELISA.** (A) Positive of nano-p-ELISA. (B) Negative of nano-p-ELISA. (**C**) Positive of tra-p-ELISA. (D) Negative of tra-p-ELISA.
(TIF)

**S1 Table. Information of the patient.**
(DOCX)

**S2 Table. The OMPs' accession numbers of *Brucella* in NCBI protein database.**
(DOC)

**S1 Data. Supporting data for Fig 1.**
(XLS)

**S2 Data. Supporting data for Fig 3.**
(XLS)

**S3 Data. Supporting data for Fig 5.**
(XLSX)

**S4 Data. Supporting data for XPS.**
(XLS)

**S5 Data. Supporting data for Fig 4A.**
(TXT)

## Acknowledgments

We thank the School of Public Health of Jilin University for their gift of the brucellosis serum samples.

## Author Contributions

**Conceptualization:** Dehui Yin, Mingjun Sun, Hai Jiang, Jingpeng Zhang.

**Data curation:** Dehui Yin, Qiongqiong Bai, Xiling Wu, Han Li.

**Formal analysis:** Dehui Yin, Qiongqiong Bai, Jihong Shao.

**Funding acquisition:** Dehui Yin.

**Investigation:** Dehui Yin, Mingjun Sun.

**Methodology:** Dehui Yin, Qiongqiong Bai, Xiling Wu, Han Li, Jihong Shao, Mingjun Sun.

**Project administration:** Dehui Yin, Mingjun Sun, Jingpeng Zhang.

**Resources:** Dehui Yin, Han Li, Mingjun Sun, Jingpeng Zhang.

**Supervision:** Dehui Yin, Mingjun Sun, Jingpeng Zhang.

**Validation:** Dehui Yin, Mingjun Sun.

**Writing – original draft:** Dehui Yin, Qiongqiong Bai.

**Writing – review & editing:** Dehui Yin, Mingjun Sun, Hai Jiang, Jingpeng Zhang.

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
