## [Decision Letter · Decision Letter 0]

1 Jun 2021

Dear Dr Yin,

Thank you very much for submitting your manuscript "Paper-based ELISA diagnosis technology for human brucellosis based on a multiepitope fusion protein" for consideration at PLOS Neglected Tropical Diseases. As with all papers reviewed by the journal, your manuscript was reviewed by members of the editorial board and by several independent reviewers. In light of the reviews (below this email), we would like to invite the resubmission of a significantly-revised version that takes into account the reviewers' comments. 

We cannot make any decision about publication until we have seen the revised manuscript and your response to the reviewers' comments. Your revised manuscript is also likely to be sent to reviewers for further evaluation.

Sincerely,

Alyssa E Barry

Associate Editor

Sergio Oliveira

Deputy Editor

Reviewer's Responses to Questions

**Key Review Criteria Required for Acceptance?**

**Methods**

-Are the objectives of the study clearly articulated with a clear testable hypothesis stated?

-Is the study design appropriate to address the stated objectives?

-Is the population clearly described and appropriate for the hypothesis being tested?

-Is the sample size sufficient to ensure adequate power to address the hypothesis being tested?

-Were correct statistical analysis used to support conclusions?

-Are there concerns about ethical or regulatory requirements being met?

Reviewer #1: Objectives and hypothesis are not clear in the manuscript. Methods need details and not matching with the results provided.

Section starting from L106 and 111 could be pooled together to make it comprehensive and easy to follow. The algorithm used to determine length and selection of sequence of peptides predicted and selected for the study are not mentioned. Also, the number of peptide epitopes predicted are missing. Were only 22 peptides predicted and used in screening by ELISA and for fusion protein preparation? Or 22 peptides were selected? The final length and details of sequence for fusion protein produced are missing in methods, as provided in table 1.

L236-Describe the positive sera used for screening of peptides? Is it the reference sera or from 121 known positive sera? Were all sera reacted to all 22 peptides?

L213-nanomodified p-ELISA (nano-p-ELISA) method comparison using traditional p-ELISA is described, however the details for nano-p-ELISA is not mentioned in the manuscript.

Sample size is sufficient and population is also clearly described. The statistical analysis support the findings. No concerns about ethical and regulatory requirements.

Reviewer #2: Minor revision

I would suggest for the authors to add the Figure on the schematic diagram of developed paper-based ELISA for human brucellosis using multiepitope fusion protein.

**Results**

-Does the analysis presented match the analysis plan?

-Are the results clearly and completely presented?

-Are the figures (Tables, Images) of sufficient quality for clarity?

Reviewer #1: Results of LPS antigen are shown in figure 3 C and D, but details of LPS antigen are missing in method section and not mentioned elsewhere in the manuscript.

L236-Describe the positive sera used for screening of peptides? Is it the reference sera or from 121 known positive sera? Were all sera reacted to all 22 peptides?

Nanomodified p-ELISA (nano-p-ELISA) method comparison using traditional p-ELISA is unclear. In discussion L365, authors also mention comparison with traditional ELISA, please provide details.

Reviewer #2: Overall the results are clearly presented accept that Figure 1. I would suggest the authors to replace with better quality of the graph to show clearly results of iELISA of each peptide identification-positive brucellosis serum

**Conclusions**

-Are the conclusions supported by the data presented?

-Are the limitations of analysis clearly described?

-Do the authors discuss how these data can be helpful to advance our understanding of the topic under study?

-Is public health relevance addressed?

Reviewer #1: This section needs to be revised with relevance to findings of the manuscript.

L340-341-should it be part of results?

L345-346- need justification for the statement for cross-reactivity.

L357-359 should be part of introduction and not discussion.

Reviewer #2: The conclusions are supported the data presented and the authors have explained the public health relevance for human brucellosis by using the developed method (paper-based ELISA on multiepitope fusion protein)

**Editorial and Data Presentation Modifications?**

Reviewer #1: Authors need to carefully review terminology used in the manuscript. The term “antigenicity” is misleading, when throughout the manuscript the fusion peptide was used as a target for detection of antibodies in human sera.

Reviewer #2: Minor revision

**Summary and General Comments**

Reviewer #1: (No Response)

Reviewer #2: 1. This study is important to be published as the developed method will help the human brucellosis diagnosis.

2. The study is novel and has shown the better performance in terms of sensitivity and specificity of the test.

3. Please include the details of the ethical approval documents.

PLOS authors have the option to publish the peer review history of their article (what does this mean?). If published, this will include your full peer review and any attached files.

Reviewer #1: Yes: Kalpana Agnihotri

Reviewer #2: No
---

## [Decision Letter · Decision Letter 1]

31 Jul 2021

Dear Dr Yin,

We are pleased to inform you that your manuscript 'Paper-based ELISA diagnosis technology for human brucellosis based on a multiepitope fusion protein' has been provisionally accepted for publication in PLOS Neglected Tropical Diseases.

Best regards,

Alyssa E Barry

Associate Editor

Sergio Oliveira

Deputy Editor

Reviewer's Responses to Questions

**Key Review Criteria Required for Acceptance?**

**Methods**

-Are the objectives of the study clearly articulated with a clear testable hypothesis stated?

-Is the study design appropriate to address the stated objectives?

-Is the population clearly described and appropriate for the hypothesis being tested?

-Is the sample size sufficient to ensure adequate power to address the hypothesis being tested?

-Were correct statistical analysis used to support conclusions?

-Are there concerns about ethical or regulatory requirements being met?

Reviewer #1: Revised methods are clear and reads well, all suggestions provided are appropriately implemented and are satisfactory.

Reviewer #2: All the important points have been addressed clearly in mothodology

**Results**

-Does the analysis presented match the analysis plan?

-Are the results clearly and completely presented?

-Are the figures (Tables, Images) of sufficient quality for clarity?

Reviewer #1: The result section is improved considerably and reads well in accordance with the methods. All suggested changes are addressed and satisfactory.

Reviewer #2: The results are sufficient.

**Conclusions**

-Are the conclusions supported by the data presented?

-Are the limitations of analysis clearly described?

-Do the authors discuss how these data can be helpful to advance our understanding of the topic under study?

-Is public health relevance addressed?

Reviewer #1: Conclusion section revised reads well.

Reviewer #2: The conclusion are acceptable and has suggested the important of newly diagnostic technology (paper-based ELISA technology) using the multiepitope fusion protein as an alternative to LPS.

**Editorial and Data Presentation Modifications?**

Reviewer #1: The revised manuscript has improved after the changes made and reads well now.

Reviewer #2: Modification needed for English editing.

**Summary and General Comments**

Reviewer #1: Satisfactory

Reviewer #2: After English editing, the manuscript is ready to be published.

PLOS authors have the option to publish the peer review history of their article (what does this mean?). If published, this will include your full peer review and any attached files.

Reviewer #1: **Yes: **Kalpana Agnihotri

Reviewer #2: No

---

## [Editor Report · Acceptance letter]

11 Aug 2021

Dear Dr Yin,

We are delighted to inform you that your manuscript, "Paper-based ELISA diagnosis technology for human brucellosis based on a multiepitope fusion protein," has been formally accepted for publication in PLOS Neglected Tropical Diseases.

Best regards,

Shaden Kamhawi

co-Editor-in-Chief

Paul Brindley

co-Editor-in-Chief
